# A mycorrhizae-like gene regulates stem cell and gametophore development in mosses

Shuanghua Wang[1,2], Yanlong Guan[1], Qia Wang[1], Jinjie Zhao[1], Guiling Sun [3], Xiangyang Hu[4], Mark P. Running[5], Hang Sun[1 ✉] & Jinling Huang [1,3,6 ✉]

Plant colonization of land has been intimately associated with mycorrhizae or mycorrhizae-like fungi. Despite the pivotal role of fungi in plant adaptation, it remains unclear whether and how gene acquisition following fungal interaction might have affected the development of land plants. Here we report a macro2 domain gene in bryophytes that is likely derived from Mucoromycota, a group that includes some mycorrhizae-like fungi found in the earliest land plants. Experimental and transcriptomic evidence suggests that this macro2 domain gene in the moss *Physcomitrella patens*, *PpMACRO2*, is important in epigenetic modification, stem cell function, cell reprogramming and other processes. Gene knockout and over-expression of *PpMACRO2* significantly change the number and size of gametophores. These findings provide insights into the role of fungal association and the ancestral gene repertoire in the early evolution of land plants.

[1] Key Laboratory for Plant Diversity and Biogeography of East Asia, Yunnan Key Laboratory for Fungal Diversity and Green Development, Kunming Institute of Botany, Chinese Academy of Sciences, Kunming 650201, China. [2] University of Chinese Academy of Sciences, Beijing 100049, China. [3] State Key Laboratory of Crop Stress Adaptation and Improvement, Key Laboratory of Plant Stress Biology, School of Life Sciences, Henan University, Kaifeng 475001, China. [4] Shanghai Key Laboratory of Bio-Energy Crops, School of Life Sciences, Shanghai University, Shanghai 200444, China. [5] Department of Biology, University of Louisville, Louisville, KY, USA. [6] Department of Biology, East Carolina University, Greenville, NC, USA. ✉email: sunhang@mail.kib.ac.cn; huangj@ecu.edu

A milestone in the evolution of plants is their conquest of land, a feat accomplished through a partnership with fungi, most commonly mycorrhizae-like association[1,2]. The dramatic habitat transition from water to land posed tremendous challenges for plants, including desiccation, increased UV irradiation and temperature fluctuation, and exposure to novel pathogens[3,4]. In response, early land plants evolved various strategies and features to adapt to hostile terrestrial environments[5,6]. In addition to their many physiological innovations, land plants underwent major developmental changes, notably three-dimensional growth as well as alternation of gametophyte and sporophyte generations[7,8]. Such developmental changes allowed a more complex body plan and morphological diversity in land plants, which would in turn enhance their reproductive success, survival, and niche expansion[7–9].

Understanding the adaptive strategies in early land plants requires a complete picture of their underlying molecular mechanisms. Bryophytes (liverworts, hornworts, and mosses) include extant members of early-diverging land plant lineages[10,11]. In addition to their important systematic position, bryophytes have retained many structures and functions, including genes and pathways, that were inherited from ancestral land plants[9,12]. Some of these genes or pathways might have been lost secondarily from other land plants over time, but they are still found in bryophytes. These ancestral genes and pathways, together with those inherited from charophytes (the closest relatives of land plants), may not only provide critical insights into the adaptive strategies of plants during their transition from water to land[5,6,13], but also the sequence of evolutionary changes that occurred when land plants became increasingly complex[14], both structurally and physiologically. Nevertheless, compared to flowering plants, bryophyte model organisms, such as the moss *Physcomitrella patens* and the liverwort *Marchantia polymorpha*, remain largely understudied[15,16]. Frequently, knowledge of the physiological and developmental processes in bryophytes heavily relies on studies on homologous genes and processes in flowering plants (e.g., *Arabidopsis thaliana*). This approach of evolutionary development has identified some core components of key pathways conserved throughout land plant evolution, but, on the other hand, has provided little information on the toolkit specific to early land plants. For instance, several genes, including *Defective Kernel 1*, *APB*-like and *CLAVATA*-like genes, are known to be conserved in stem cell functions and three-dimensional growth between bryophytes and flowering plants[17–19]. Nevertheless, how other ancestral genes or pathway components (e.g., those only retained in basal land plants) contributed to stem cell development and three-dimensional growth of early land plants remains obscure.

Macrodomains are known for their capability of binding or cleaving ADP-ribose from cellular molecules and, therefore, play a key role in ADP-ribosylation, an important post-translation modification mechanism that is yet to be fully elucidated[20–22]. They regulate a number of cellular activities, such as chromatin modification, transcription and translation, DNA repair, and cell differentiation[20,22]. Within the macrodomain superfamily, the macro2 family (Pfam 14519) is the least studied and its biological functions are largely unclear[20]. In this study, we report a macro2 domain gene in bryophytes that was likely acquired by the ancestral land plant from mycorrhizae-like fungi. This mycorrhizae-like macro2 domain gene has been lost secondarily from vascular plants but is still retained in bryophytes. Experimental evidence shows that the macro2 domain gene in *P. patens* (*PpMACRO2*) regulates multiple key processes, including the development of stem cells and gametophores (the dominant form of three-dimensional growth in bryophytes), cell reprogramming and tissue regeneration, as well as epigenetic changes. We

speculate that *PpMACRO2* is involved in histone modification through ADP-ribosylation, which in turn triggers additional epigenetic changes through other mechanisms in *P. patens*. We further discuss the adaptive role of the macro2 domain and ancestral gene repertoire in land plants.

## Results

**Mycorrhizae-like fungal origin of land plant macro2 gene**. The PpMACRO2 protein includes a single macro2 domain of 145 aa, with an additional N-terminus of 97 aa and a short C-terminus of about 64 aa. With the PpMACRO2 protein sequence (Genbank accession number: XP_024388278) as query, we performed a BLAST search of the NCBI non-redundant (*nr*) protein sequence database, the 1000 plants project (OneKP) and other resources, including the recently published genomes of hornworts (*Anthoceros angustus*), ferns (*Azolla filiculoides* and *Salvinia cucullata*) and charophytes (e.g., *Chara braunii*, *Spirogloea muscicola*, *Mesotaenium endlicherianum*, *Mesostigma viride*, and *Chlorokybus atmophyticus*)[23–27], as well as our internal draft genome of another charophyte, *Interfilum paradoxum*. Additional pHMMER search was performed against Reference Proteomes. Our searches provided hits only from mosses, liverworts, fungi, bacteria, viruses, and a few other eukaryotes (E-value cutoff=1e-6) (Supplementary Figs. 1 and 2). Many of the bacterial hits are annotated as phage tail proteins. No hits could be detected in hornworts and vascular plants in our analyses. Additionally, although a BLAST search of OneKP and NCBI dbEST databases, which only contain transcriptomic data, yielded hits from green algal species, no hit could be identified from any complete genome of green algae that was generated from axenic cultures. Similarly, although hits were found from the charophyte *Spirogyra pratensis* in NCBI dbEST database, our polymerase chain reaction (PCR) from genomic DNA failed to amplify sequences from an unspecified congeneric species *Spirogyra sp.* (Supplementary Fig. 3 and Supplementary Table 1). Therefore, the possibility that these green algal hits are from foreign sources (i.e., contamination) cannot be excluded.

Notably, moss and liverwort macro2 protein sequences not only had the highest coverage (up to 78%) and percent identity (up to 59%) with hits from Mucoromycota fungi (*Rhizophagus irregularis*, *Gigaspora rosea*, *Jimgerdemannia flammicorona*, etc.), they also uniquely shared several amino acid residues (Fig. 1a and Supplementary Fig. 5a). Consistent with the sequence comparison, phylogenetic analyses showed that PpMACRO2 and its homologs from other mosses, liverworts and fungi (mostly Mucoromycota fungi) formed a well-supported clade, which in turn grouped with other fungal sequences (Supplementary Figs. 4 and 5b).

**PpMACRO2 protein is localized in both nucleus and cytoplasm**. The *PpMACRO2* gene is located on chromosome 11 of *P. patens* and annotated to include 1499 nucleotides according to Phytozome (phytozome.jgi.doe.gov/pz/portal.html). Although the annotated *PpMACRO2* gene consists of three introns and four exons, our reverse transcription PCR (RT-PCR) experiments were only able to recover exons 1, 3, and 4, without the annotated exon 2 (Fig. 1b). This result is consistent with the Phytozome annotation that the transcription level of exon 2 is minimal (Supplementary Fig. 6). The RT-PCR cloned sequence is predicted to contain an intact macro2 domain (Supplementary Fig. 7).

To visualize the subcellular localization of PpMACRO2, we performed transient protoplast transformation of the *PpMACRO2* gene. The coding sequence of *PpMACRO2* was tagged with enhanced green fluorescent protein (EGFP) and transformed using polyethylene glycol (PEG)-mediated protoplast transient

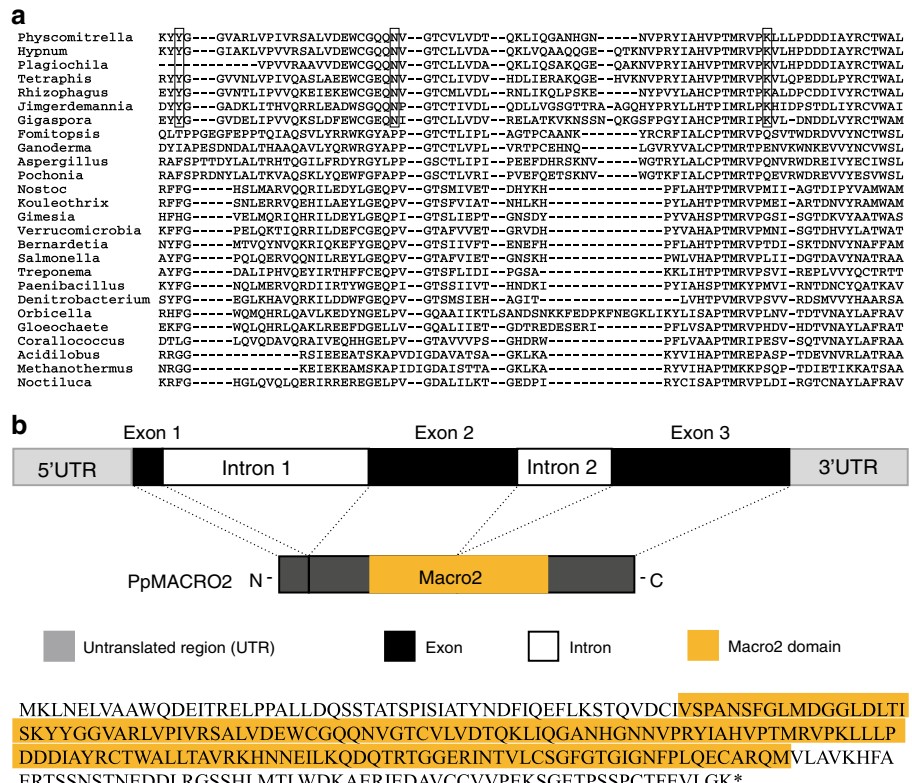

**Fig. 1 PpMACRO2 sequence relationship and gene annotation. a** Multiple alignment for PpMACRO2 protein sequence and homologs sampled from different major groups. Boxes in the alignment show amino acids uniquely shared by bryophytes and Mucoromycota fungi. Detailed molecular phylogeny of PpMACRO2 and homologs is shown in Supplementary Figs. 4 and 5. **b** Schematic diagram of PpMACRO2 gene structure and encoded protein sequence based on RT-PCR and sequencing evidence. Exons 1, 2, and 3 correspond to exons 1, 3, and 4, respectively, of the P. patens v3.3 annotation in Phytozome.

expression in P. patens. GFP fluorescence was observed in both the nucleus and cytoplasm (Supplementary Fig. 8).

**PpMACRO2 promotes protonema growth and gametophore budding.** To understand the functions of PpMACRO2, we generated four knockout (ko) and three over-expression (OE) genetically modified plants of PpMACRO2, respectively. The ko and OE plants were characterized by genotyping the transformants using genomic PCR, real-time quantitative RT-PCR (qRT-PCR), and Southern blotting (Supplementary Figs. 9 and 10). All ko and OE plants were haploids based on flow cytometric analyses (Supplementary Fig. 11). The wild-type (WT), ko, and OE plants were cultivated under normal growth conditions, and their phenotypes were observed and measured. Compared to WT plants of P. patens, the ko mutants produced smaller protonemata, but larger and fewer gametophores; the reverse was true for OE lines (Figs. 2 and 3a–d). Specifically, the average number of gametophores in ko mutants decreased by about 29% relative to WT plants of P. patens, whereas the number of gametophytes increased by 33–71% in the three different OE lines (Fig. 3c). Furthermore, the ko mutants also produced longer gametophores (Fig. 2d).

To further understand the role of PpMACRO2 in gametophore development, we generated transgenic plants of P. patens in which EGFP and β-glucuronidase (GUS) reporter genes were inserted inframe immediately before the stop codon of PpMA-CRO2 via homologous recombination (Supplementary Fig. 12). The resulting plants expressed a PpMACRO2-EGFP-GUS fusion protein under the control of its native promoter within the endogenous genomic environment. The PpMACRO2pro: PpMACRO2-EGFP-GUS lines had no visible developmental

differences compared to WT plants. Fluorescent signal of the fusion protein was detected during the development of gametophores in P. patens. Notably, the signal was particularly strong in developing buds and apical meristems of leafy gametophores (Fig. 3e, f and Supplementary Fig. 13). This evidence, together with changes in the number and size of gametophores in ko and OE plants (Figs. 2 and 3a–d), suggests that PpMACRO2 plays a key role in gametophore development of P. patens.

It has been known that cytokinin induces gametophore bud formation, and over-budding can be obtained by exogenous application of cytokinin[28,29]. To investigate the interplay between PpMACRO2 and cytokinin in bud formation, we treated P. patens WT plants and PpMACRO2 ko mutants with $1 \mu mol \, l^{-1}$ 6-benzylaminopurine (BA), a synthetic cytokinin. Under normal growth conditions, the ko mutants generated about 30% fewer gametophores than WT plants. Under 6-BA treatment, the number of gametophores in the ko mutants, however, was only about 12% fewer compared with the wild type (Fig. 3d), indicating that cytokinin could partially rescue PpMACRO2 ko mutants in gametophore budding.

**PpMACRO2 in stem cell development and cell reprogramming.** To assess whether PpMACRO2 participates in other developmental processes in P. patens, we performed GUS histochemical assays using PpMACRO2pro:PpMACRO2-EGFP-GUS. GUS expression was consistently detected in stem cells throughout the lifecycle of P. patens (Fig. 4). Upon germination of spores and throughout the protonema phase, GUS staining was detected in both chloronemata and caulonemata, but clearly stronger in apical and side-branch stem cells (Fig. 4a–c). During the development of gametophores, strong GUS staining was

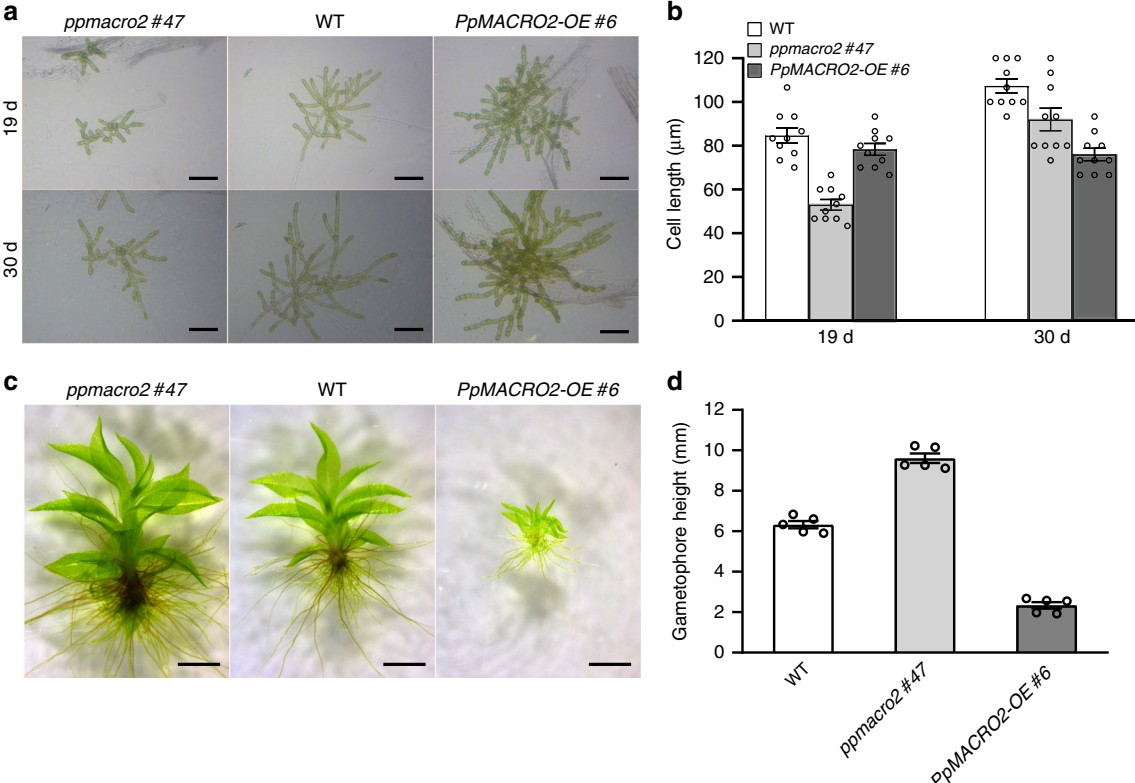

**Fig. 2 _PpMACRO2_ affects sizes of protonemata and gametophores in _P. patens_. a, b** Protonemal cell growth and elongation in WT, _ko_ and _OE_ plants. Protonemal cell length in _ko_ mutants decreased by about 40% than in the wild type after 19 days, but there was no significant difference between _OE_ lines and the wild type. After 30 days, protonemal cells in _ko_ mutants were only about 15% shorter compared to the wild type, whereas protonemal cells of _OE_ lines were about 30% shorter. These results show that protonemata grow much faster in _ko_ mutants but slower in _OE_ lines compared to the wild type between day 19 and day 30. Micrograph images shown were observed from at least three biological replicates. Data show means ± s.e.m. of ten biological replicates for WT and mutant plants. **c, d** Sizes of gametophores developed from four-week-old WT, _ko_, and _OE_ plants. Gametophore heights were measured twice independently, with similar results. Micrograph images provided were observed from five biological replicates. Data show means ± s.e.m. of five biological replicates. Scale bar: 200 μm in **a**; 500 μm in **c**.

mostly restricted to the three-faced bud apical cells and leafy apical stem cells (Fig. 4d, e). Before fertilization, GUS activity was detected in developing archegonia (Fig. 4f), where staining was stronger in unfertilized eggs, whereas no GUS staining was detected in antheridia. After fertilization, GUS signal was detectable in mature sporophytes and individual spores (Fig. 4g, h). The above observation suggests that _PpMACRO2_ is involved in stem cell development in _P. patens_.

Given the strong expression of _PpMACRO2_ in apical stem cells of both protonemata and gametophores, we investigated whether and how _PpMACRO2_ might affect cell reprogramming and regeneration. Protoplast regeneration was first performed for WT, _ko_ and _OE_ plants, respectively. During the first two days after protoplast isolation, cells divided more quickly in _OE_ lines than in WT and _ko_ plants (Fig. 5a). After five days, more cells formed in _OE_ lines compared to the wild type, whereas cells were fewer and shorter in _ko_ mutants, consistent with the smaller protonemata observed in the early propagation stage of _ko_ mutants (Fig. 2a, b). Between day 5 and day 10, branching of protonemata was faster in _OE_ lines, whereas it was slower in _ko_ mutants, relative to WT plants. On day 13, we observed bud initiation in _OE_ lines, but not in WT and _ko_ plants (Fig. 5a). Protonema branching became faster and protonema cells were longer in _ko_ mutants after 13 days, but fewer gametophores were produced compared to WT and _OE_ plants.

_Physcomitrella patens_ tissues are capable of regeneration when damaged[30]. To understand the role of _PpMACRO2_ in tissue

regeneration, we detached leaves from gametophores of WT, _ko_ and _OE_ plants, respectively, and cultivated them on BCD medium. Approximately 70 h after detachment, protonema filaments began to emerge in WT plants. During the same growth period, more filaments were observed from the _OE_ lines, whereas none was formed in the _ko_ mutants (Fig. 5b). To visualize the expression of _PpMACRO2_ during tissue regeneration, we performed parallel experiments using _PpMACRO2-EGFP-GUS_ knockin lines and examined GUS staining and GFP fluorescence on the detached leaves (Fig. 5c and Supplementary Fig. 14). The GUS expression pattern in WT plants of _P. patens_ was similar to that previously reported for _PpFIE_, which encodes a component of Polycomb group complex in stem cell maintenance[31]. At 48 h after detachment, GUS staining and GFP fluorescence became visible on the surface of detached leaves (Fig. 5c). Protonema filaments then developed gradually from the GUS-stained and GFP-fluoresced cells. These data indicate that _PpMACRO2_ promotes cell reprogramming and tissue regeneration in _P. patens_.

**_PpMACRO2_ affects epigenetics and transcription factors.** Stem cell and gametophore development involves a number of activities[29,32]. To understand how _PpMACRO2_ may affect other activities, we generated RNA-seq data for both _ko_ and _OE_ lines of _PpMACRO2_, and then identified differentially expressed genes relative to the WT plants. Notably among the differentially

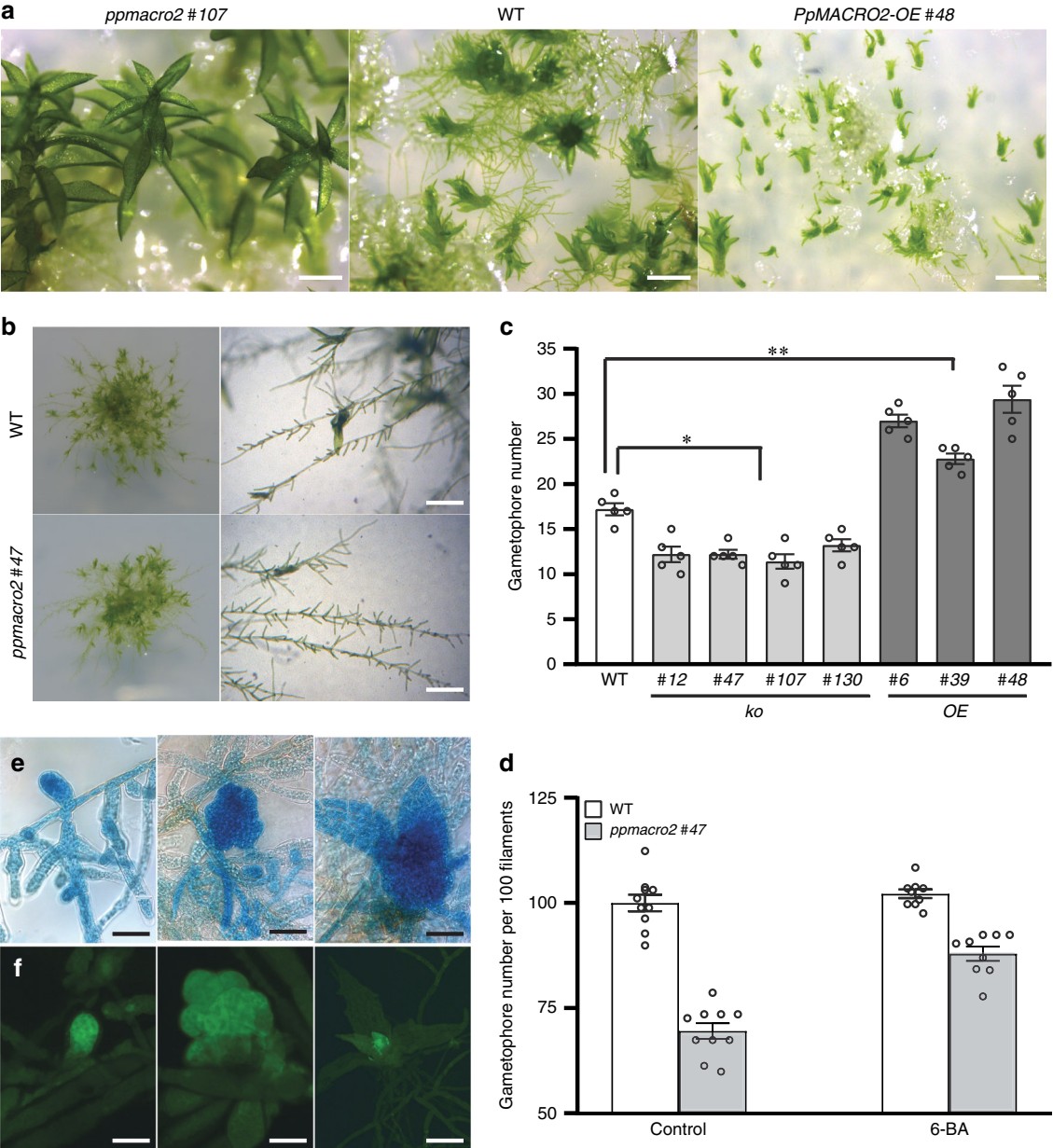

**Fig. 3 *PpMACRO2* affects the number of gametophores developed in *P. patens*. a, c** Number of gametophores developed from four-week-old WT, *ko* and *OE* plants. Seven-day-old protonemata of WT, *ko* and *OE* plants were harvested and suspended with sterile water, respectively, and then crushed. The same amount of suspension ($OD_{600} = 0.4$) was propagated onto BCD medium and grown for about four weeks. Gametophore numbers were counted under microscope's field of view. Micrograph images given were observed from five biological replicates. Data show means ± s.e.m. of five biological replicates for WT and mutant plants. Asterisks indicate a statistically significant difference compared with WT plants based on a two-tailed Student's *t*-test (*$p < 0.05$, **$p < 0.01$). **b, d** Number of gametophores per filament in WT and *ko* plants of *PpMACRO2*. Protonemata of approximately 1 mm in diameter were transplanted onto BCD medium from the wild type and *ppmacro2* #47 respectively, and cultivated for three weeks. Ten clones were counted for the wild type and *ko* mutant, respectively. Under normal growth conditions, the *ko* mutant generated about 30% fewer gametophores than the wild type. However, under $1 \mu mol\, l^{-1}$ 6-BA treatment, the number of gametophores in the *ko* mutant was only about 12% fewer compared to the wild type. Micrograph images provided were observed from ten biological replicates. Data show means ± s.e.m. of ten biological replicates. **e, f** GUS staining and green fluorescence of gametophore development. Left to right show a swollen gametophore apical cell, a gametophore with multiple cells, and a leafy gametophore, respectively. Micrograph images shown were observed from at least three biological replicates. Scale bars: 500 μm in **a**, **b**; 50 μm in **e**, **f**.

expressed genes are those related to epigenetic modification (Fig. 6 and Supplementary Fig. 15). These genes encode at least four SET domain proteins or putative histone-lysine methyltransferases (*Pp3c13_4470*, *Pp3c13_19810*, *Pp3c1_8530*, and *Pp3c17_14770*), a homolog of Sin-associated protein 30 (SAP30) (*Pp3c20_6230*), and a homolog of methyl-CpG binding domain-containing protein 9 (MBD9) (*Pp3c5_19640*). In line with the transcriptomic data, our searches of the STRING database predicted protein-protein interaction of PpMACRO2 with histones H2A and H2B (Supplementary Fig. 16)[33]. Furthermore, several families of developmental transcription factors were also differentially expressed in *PpMACRO2 ko* and *OE* lines. In particular, *AP2* and homeobox genes were down-regulated in *ko* mutants and, conversely, up-regulated in *OE* lines (Fig. 6a). These

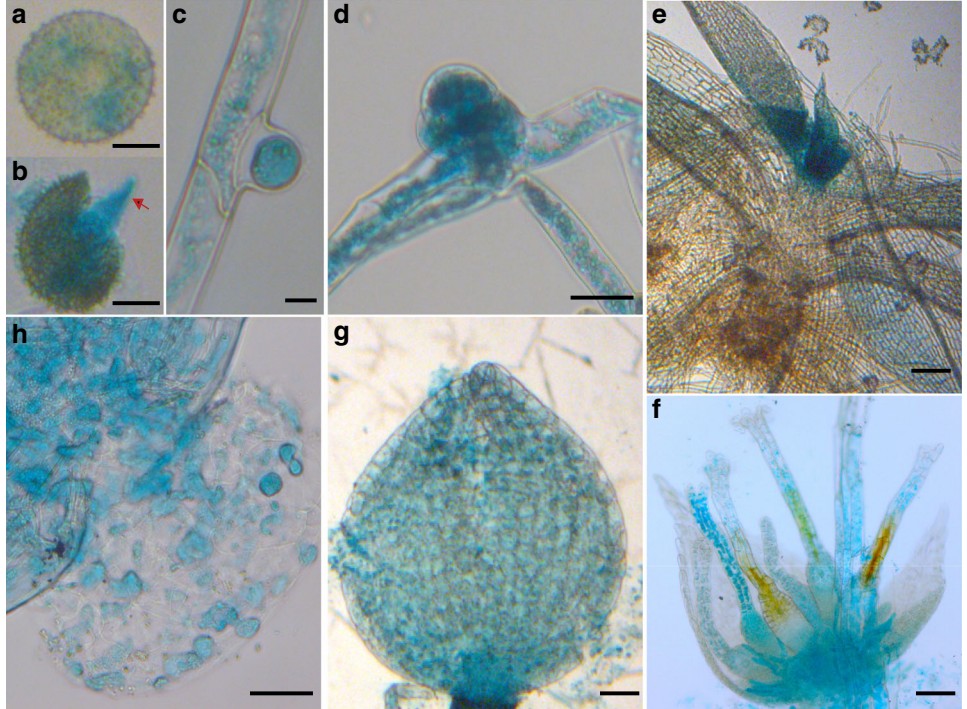

**Fig. 4 Expression profile of *PpMACRO2* determined by histochemical GUS assays. a** Spores. **b** Germinating spores, with the red arrow indicating the chloronema apical stem cell. **c** Protonema side-branch apical stem cell. **d** Juvenile gametophore. **e** Leafy gametophore. **f** Developing archegonium, with GUS activity detected in the egg cell as well as in the archegonia tissue. **g** Sporangium. **h** Individual spores within sporangium. Micrograph images provided were observed from at least three biological replicates. Scale bars: 25 μm in **a**, **b**; 100 μm in **c**, **d**, **f**–**h**; 500 μm in **e**.

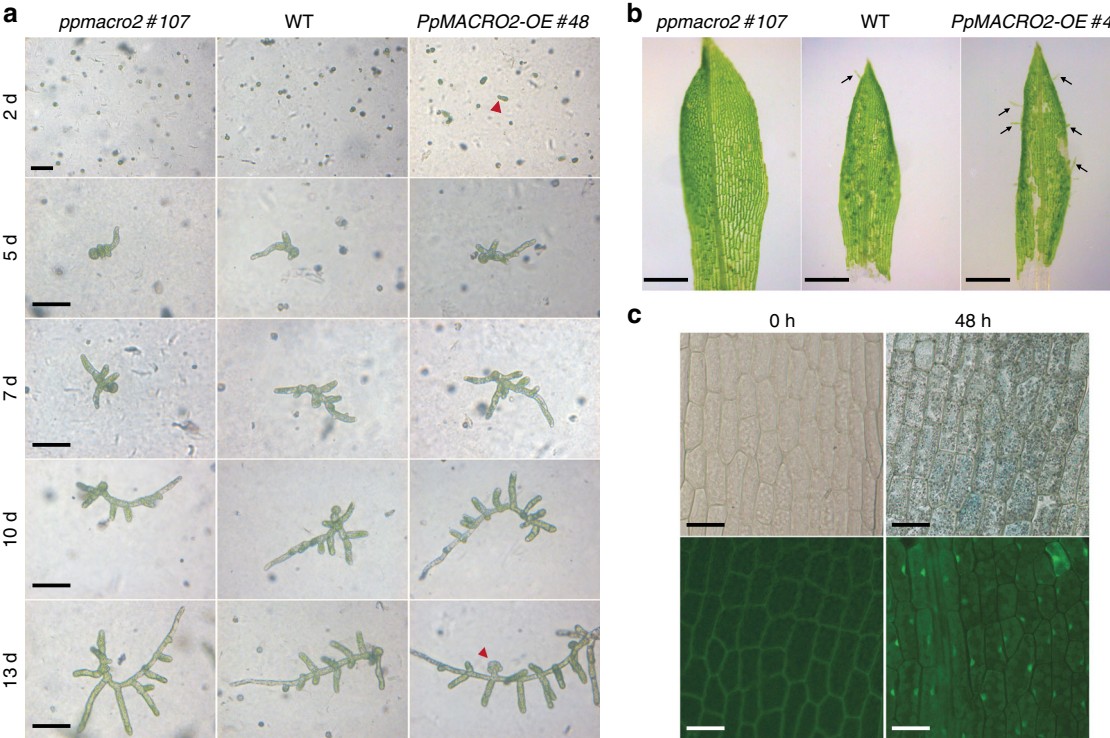

**Fig. 5 *PpMACRO2* promotes cell reprogramming in *P. patens*. a** Protoplast regeneration, with red arrowheads indicating differentiated cells and buds. **b** Tissue regeneration using detached leaves from WT plants, as well as *ko* and *OE* mutants of *P. patens*. Black arrows indicate emerged protonema filaments. **c** GUS staining and GFP fluorescence of leaves were observed from *PpMACRO2-EGFP-GUS* transgenic lines after being detached for 48 h. Micrograph images shown were observed from five biological replicates. Scale bars: 100 μm in **a**, **c**; 500 μm in **b**.

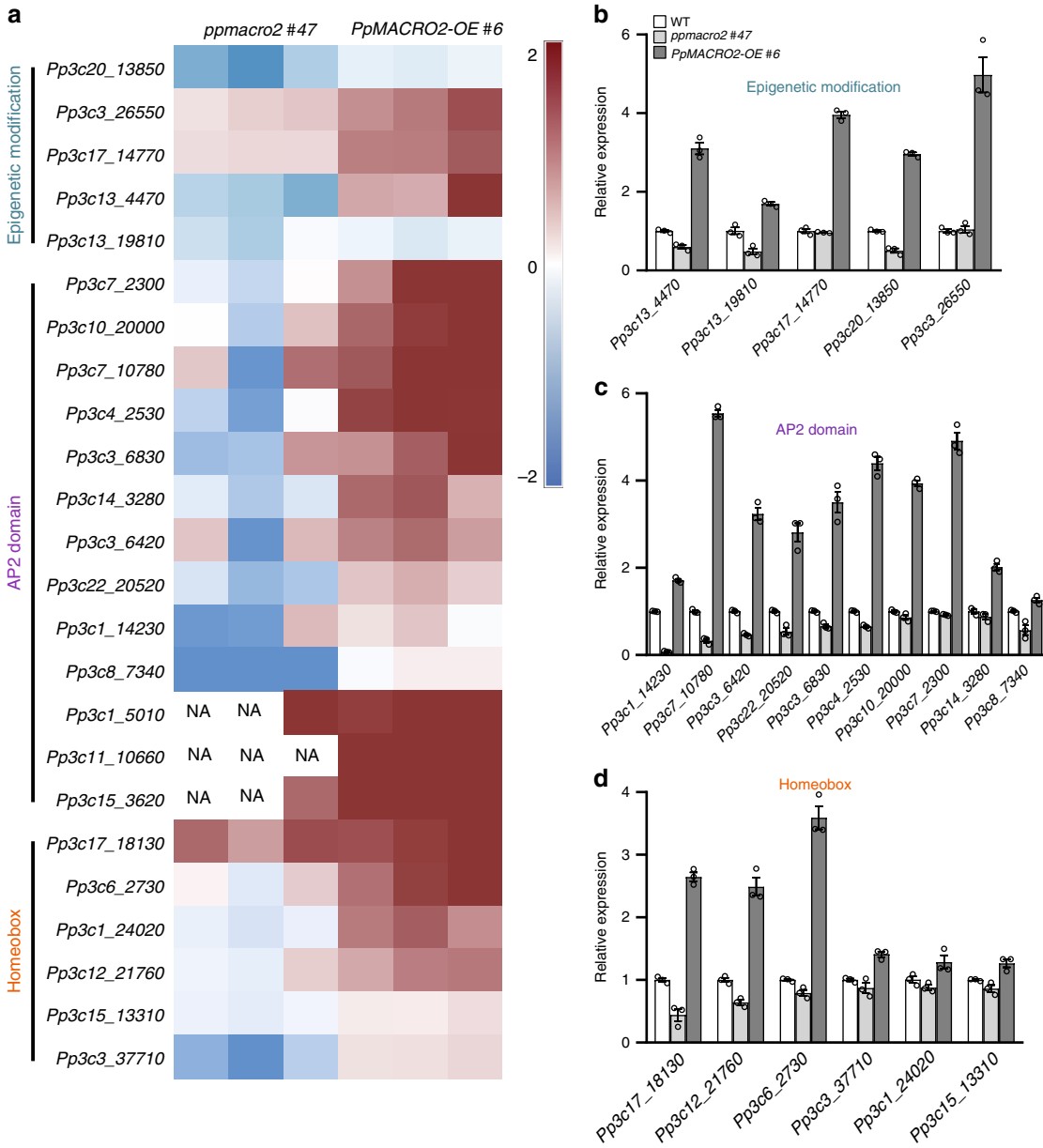

**Fig. 6 Differentially expressed genes related to epigenetic modification and stem cell development in *ko* and *OE* lines of *P. patens*. a** Transcriptional profile from RNA-seq data for a subset of genes related to epigenetic modification and developmental transcription factors in WT plants, as well as *ko* and *OE* mutants. Total RNA extracted from WT plants and mutants was used for RNA-seq analyses. **b–d** Transcript abundance of the subjected genes was confirmed through qRT-PCR with three independent biological replicates, normalized to *PpEF1a*. Phytozome identifier is used for each gene, and additional information for each gene is provided in Supplementary Table 3. Data show means ± s.e.m.

transcription factors are often linked to stem cell development and cell reprogramming[34]. For instance, AP2-type and AP2-like transcription factors not only determine stem cell identity and gametophore formation, but also induce cell reprogramming in *P. patens*[17,35]. Homeobox genes are essential for the development of apical meristems in plants[36,37]. Other noteworthy and differentially expressed genes in *ko* and *OE* plants included those related to cell wall formation (e.g., xyloglucan endo-transglycosylase, pectinesterase) and cell division (kinesin-like protein, tubulin beta, and protein regulator of cytokinesis 1), which were often down-regulated in both *ko* and *OE* lines (Supplementary Table 2).

To validate the expression pattern from RNA-seq data, we performed qRT-PCR experiments on a subset of the above genes, particularly the genes related to epigenetic modification and developmental transcription factors. RNA-seq expression data for

a majority of selected genes were confirmed (Fig. 6b–d and Supplementary Table 3). These data support the role of *PpMACRO2* in epigenetic modification, stem cell development, and related processes.

## Discussion

Information on ancestral physiological and developmental pathways in early land plants provides unique insights into the strategies or toolkits adopted by plants during their transition from water to land. Bryophytes retain many features evolved in early land plants and thus provide a living laboratory to understand these pathways and related adaptive strategies[9,12]. In this study, we report a macrodomain gene, *PpMACRO2*, that is distributed among bryophytes but absent from vascular plants. We show that

*PpMACRO2* is important in stem cell development, cell reprogramming, protonema development, gametophore budding and other processes in *P. patens*. Considering its universal expression throughout major developmental phases in *P. patens* and its presence in many other bryophytes, it is likely that *PpMACRO2* played a fundamental role in the cellular activity and gametophore development during the early evolution of land plants.

Macrodomains are structurally conserved and often functionally related to ADP-ribosylation[20,38]. Because of their distribution in viruses and all three major domains of life (bacteria, archaea, and eukaryotes), macrodomains are believed to be ancient and critical in miscellaneous cellular processes, though the number of macrodomains identified thus far remains limited[22]. Most of the identified macrodomain families (or classes) are well correlated with their functional roles, but little is known about the macro2 family[20]. In plants, although several macrodomain-containing genes have been reported in *A. thaliana*[20,39], only two poly(ADP-ribose) glycohydrolase genes (*PARGs*), which are structurally conserved among eukaryotes, have been functionally investigated[40–42]. To our knowledge, no macro2 domain gene has been reported in plants thus far.

Our data provide a glimpse into the functions and mechanisms of the macro2 domain in the development of plants and other eukaryotes. In particular, both transcriptomic and qRT-PCR data indicate that several key genes in histone modification are differentially regulated in *PpMACRO2 ko* and *OE* lines (Fig. 6a, b and Supplementary Fig. 15). This evidence strongly suggests that *PpMACRO2* is involved in chromatin changes in *P. patens*. Intriguingly, the above genes are major players of different histone modification mechanisms. For instance, SET domain proteins commonly regulate histone methylation[43–45], whereas SAP30 is a key component of the Sin3-histone deacetylase (HDAC) complex that regulates histone deacetylation[46,47]. MBD9, on the other hand, has been demonstrated in *A. thaliana* to modulate development by modifying chromatin structure via histone acetylation and DNA methylation[48]. It is noteworthy here that macrodomain-containing proteins, such as macroH2A1, macroH2A2, amplified in liver cancer 1 (ALC1) and poly(ADP-ribose) polymerases (PARPs), are known to induce chromatin conformation changes either as a histone component or a chromatin regulator[20,38,49]. We therefore propose that macro2, similar to some other macrodomain families (e.g., macroH2A-like and ALC1-like), is involved in chromatin modification through ADP-ribosylation, which in turn triggers a cascade of additional chromatin changes via other epigenetic mechanisms (e.g., histone methylation and acetylation). These cross-talks between different mechanisms may fine-tune the regulation of chromatin conformation and, therefore, the activation/repression of downstream transcription factors in stem cell development (e.g., AP2 and homeobox genes) and other processes (Fig. 7)[50,51]. This model is consistent with the observation that *PpMACRO2* is expressed in the nucleus in *P. patens* and the fact that *PpMACRO2* has a similar expression profile as the *PpFIE* gene of the Polycomb group complex (Supplementary Figs. 8 and 14), a known player in histone methylation[31,52].

Given the role of *PpMACRO2* in some key developmental processes of *P. patens*, it is puzzling that this gene has apparently been lost in vascular plants. Presumably, such selective retention in bryophytes points to a unique role of the macro2 domain during the transition of green plants from water to terrestrial environments. It is well known that ADP-ribosylation plays a crucial role in plant responses to biotic and abiotic stresses[53,54]. For instance, PARPs and the macrodomain-containing PARGs are involved in stress responses to pathogens and genotoxic agents[40,41]. Conceivably, because of the involvement of macrodomains in ADP-ribosylation, *PpMACRO2* (and its homologs in

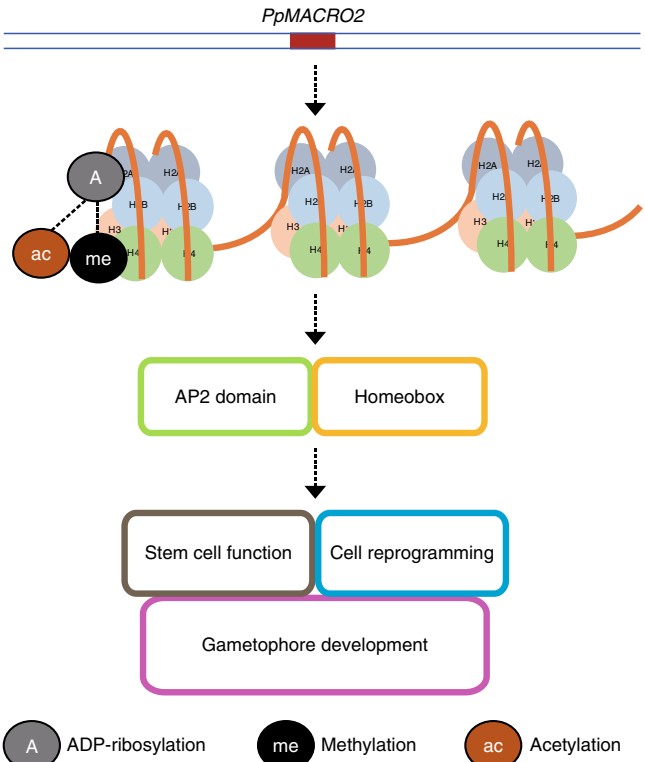

**Fig. 7 A proposed model for *PpMACRO2* in chromatin modification and development of *P. patens*.** *PpMACRO2* is likely involved in chromatin modification through ADP-ribosylation, which in turn triggers a cascade of additional chromatin changes via other epigenetic mechanisms, including histone methylation and acetylation. These epigenetic modification mechanisms may fine-tune chromatin conformation, which consequently activates/represses downstream developmental transcription factors (e.g., AP2 and homeobox) and affects stem cell function, cell reprogramming, gametophore development, and other biological processes. Dashed lines indicate that the relationship is less certain.

other bryophytes) might have facilitated the adaptation of early land plants to primeval terrestrial environments (e.g., DNA repair, pathogen resistance), in addition to their developmental role. The retention of this gene may be important for bryophytes, considering their lack of sophisticated protective mechanisms compared to other land plants. In fact, retention of genes in bryophytes (or loss of genes from vascular or seed plants) has been documented in several other cases[55–57], most of which, to a certain extent, are also related to adaptation of plants to environmental stresses. For instance, actinoporin and hemerythrin genes are involved in desiccation tolerance in mosses, but both appear to have been lost from seed plants[55,57]. ENA ATPase genes, which reportedly confer salt resistance to bryophytes, are also lost from vascular plants[56]. On the other hand, the concurrent loss of *PpMACRO2* and other stresses-related genes was also possibly associated with a shift, either structural or physiological, over the course of land plant evolution. It remains to be investigated whether this shift occurred in seedless vascular plants or seed plants.

Other than land plants and fungi, PpMACRO2 homologs are also found in bacteria, viruses, and some other eukaryotes. In bacteria, these sequences are often annotated as phage tail proteins, suggestive of a viral origin (i.e., prophages or remnant phages). Because viruses are commonly found in green algae[58–60], it is unclear whether the hits in green algal datasets were actually from associated viruses. Intriguingly, both sequence comparison

and molecular phylogeny showed that PpMACRO2 and other land plant homologs are closely related to sequences from Mucoromycota, a group that includes not only some common mycorrhizal fungi in vascular plants (e.g., *Rhizophagus irregularis*), but also mycorrhizae-like fungi found in the earliest land plants[2,61] (Supplementary Figs. 4 and 5). This sequence relatedness between land plants and Mucoromycota might have resulted from several scenarios, including differential gene losses and organellar origin. On the other hand, it is also likely that *PpMACRO2* was acquired from mycorrhizae-like fungi by the common ancestor of land plants[62]. This second scenario is not only consistent with the common belief that physical association, such as symbiosis, often facilitates horizontal gene transfer (HGT), but also supported by the amino acid residues uniquely shared by Mucoromycota and bryophyte sequences. Thus far, HGT in eukaryotes remains hotly debated[63,64] but has been reported in many eukaryotic lineages[65–67], and foreign genes have also been documented in bryophytes and their charophyte relatives[6,25,27,68]. The vast majority of the documented foreign genes in bryophytes and charophytes, however, have not been investigated experimentally. It merits further detailed studies to understand how organismal interactions and ensuing gene transfer might have impacted the abilities of green plants to colonize land.

## Methods

**Phylogenetic and protein-protein interaction analyses.** The PpMACRO2 protein sequence (NCBI accession number XP_024388278; Phytozome identifier Pp3c11_23270) was used as query to perform BLASTP searches against NCBI *nr* protein sequence database, OneKP, the marine microbial eukaryote transcriptome sequencing project (MMETSP), and other relevant databases (e.g., Phytozome, NCBI dbEST, FernBase) (E-value cutoff = 1e-6). Additional pHMMER searches were performed against References Proteomes (E-value cutoff = 1e-6). Representative protein sequences from different lineages were sampled for phylogenetic analyses. Multiple protein sequence alignments were performed using MUSCLE with manual refinement. Gaps and ambiguously aligned sites were removed from alignments. Phylogenetic analyses were performed with a maximum likelihood method using PhyML 3.1 and a distance method using neighbor of PHYLIP 3.695. ModelGenerator was used to determine the optimal model of protein substitution and rate heterogeneity. Bootstrap analyses were performed using 100 replicates.

Protein-protein interaction network analyses for PpMACRO2 were performed using STRING database (https://string-db.org).

**Plant materials and culture conditions.** The 'Gransden 2004' of *P. patens* was used as WT strain and cultured on BCD and BCDAT media at 25 °C under 16-h light and 8-h dark regime, light intensity 80 µmol photons $m^{-2}\,s^{-1}$. Protonemata of *P. patens* were grown on BCDAT medium, and gametophores were grown on BCD medium (minus ammonium tartrate).

To count the number of gametophores, 7-day-old protonemata of WT, *ko* and *OE* plants were harvested and suspended in 1.5–2 ml of sterile water, respectively, and then crushed. 1 ml of the suspension ($OD_{600} = 0.4$) was propagated onto BCD medium and grown for about four weeks. Gametophore numbers were counted under microscope's field of view. In addition, protonemata of approximately 1 mm in diameter were transplanted onto BCD medium and cultivated for 3–4 weeks. The number of gametophores was determined by counting gametophores per filament of clone. To obtain sporophytes, protonemata were planted into Jiffy7 (peat moss pot: Jiffy Products International AS, Kristansand, Norway) to grow healthy gametophores. The peat moss pots containing gametophores were submerged in water and moved to 15 °C under short-day photoperiod (8-hour light and 16-hour dark). Sporophytes began to develop after about 3 weeks of induction under low light and low temperatures. Mature sporangia were collected into a 1.5 ml microtube and sterilized using 10% Antiformin for 5 min, and then washed 3 times using sterilized water. Finally, sporangia were crushed in 1 ml sterilized water using the tip of a pipette, and the spore suspension was then mixed gently and poured onto BCD medium, and grown at 25 °C under 16-hour light and 8-hour dark photoperiod. The germinated spores were observed and photographed after 3 days of incubation.

**Plasmid construction for knockout and over-expression.** The vector pTN182 was used to delete *PpMACRO2* in WT plants of *P. patens*. Genomic fragments containing the upstream (1007 bp) and the downstream (1270 bp) flanking regions of *PpMACRO2* were inserted into the pTN182 vector, respectively. Primers used for plasmid construction are provided in Supplementary Table 4.

The vector pPOG1 was used for over-expression plasmid construction. The complete coding region of *PpMACRO2* was amplified from cDNA by RT-PCR using primers based on *P. patens* v3.3 annotation (https://phytozome.jgi.doe.gov/pz/portal.html#!info?alias=Org_Ppatens) and was subsequently cloned and sequenced. The resulting *PpMACRO2* CDS contained 857 bp spliced from 3 exons. CDS of *PpMACRO2* was amplified using primers shown in Supplementary Table 4, and cloned into the pPOG1 vector.

**Protoplast transformation.** Transformation was performed using the PEG-mediated method[69]. $1.6 \times 10^6$ protoplasts per ml were incubated with about 30 µg linearized DNA under PEG treatment, and stable lines were selected by two successive cycles of incubation on nonselective media and selective media (on media containing 20 µg $ml^{-1}$ G418 or 20 µg $ml^{-1}$ hygromycin). The stable lines were screened using genomic PCR and qRT–PCR. Moreover, Southern blotting was performed to confirm single integration in *ko* and *OE* lines. All transgenic lines were examined to be haploid with flow cytometry.

**Protein subcellular localization.** The vector pM999 was used for plasmid construction of transient expression of *PpMACRO2*. CDS of *PpMACRO2* was amplified by RT-PCR with primers given in Supplementary Table 4 and cloned into the EcoRI-SacI site of pM999, and EGFP was fused to *PpMACRO2* for transient expression. Transient transformation was performed using the PEG-mediated method. The transformed protoplasts were incubated in darkness at 25 °C for 16 h. In addition, the vector pTN85 was used to obtain *PpMACRO2*pro:*PpMACRO2-EGFP-GUS* stable lines. Genomic fragments containing the upstream (1016 bp) and the downstream (1071 bp) flanking regions of *PpMACRO2* were inserted into the pTN85 vector, respectively (Supplementary Fig. 12). Primers used for plasmid construction are provided in Supplementary Table 4. Protoplast transformation was performed and stable lines were screened. Detached leaves of *PpMACRO2*pro: *PpMACRO2-EGFP-GUS* lines were cultured on BCD medium for 48 h.

GFP signals were observed for protoplasts and detached leaves. Images were obtained using a microscope (Leica DM5500 B, Germany). The green fluorescence excitation was performed with a 488 nm Argon laser.

**Quantitative RT-PCR.** Total RNA was extracted from protonemata of *P. patens* using a RNeasy Plant Mini Kit (Qiagen) and DNase I (Solarbio). cDNA was synthesized using M-MLV reverse transcriptase (Promega M1701). qRT–PCR was performed using Bio-Rad CFX96 Real-Time System (Thermo Fisher Scientific) and TransStart Top Green qRT–PCR kit (Transgen Biotech), with three independent biological replicates. *PpEF1α* (elongation factor 1-alpha, *Phypa_439314*) was used as reference gene to calculate the relative expression.

**Genotyping of plant materials.** All genetic materials were confirmed using genomic PCR, and the primer sequences are shown in Supplementary Table 5. Products of genomic PCR were detected using gel electrophoresis. qRT–PCR was performed to detect the expression level of *ko* and *OE* lines. Primer sequences for qRT-PCR are provided in Supplementary Table 6. In addition, Southern blotting was performed to confirm single integration in *ko* and *OE* lines using primers provided in Supplementary Table 7.

**Southern blotting.** Southern blotting was performed as follows: ~3 µg of genomic DNA was digested with restriction enzyme NdeI or BglII (see Supplementary Figs. 9 and 10), run on 0.8% (w/v) agarose gel (TransGen Biotech), and transferred to a Hybond-N + nylon membrane (GE Healthcare). Probe labeling, hybridization and detection were performed using Dig High Prime DNA Labeling and Detection Starter Kit II (Roche) according to the supplier's instructions. Primers used for probe amplification are given in Supplementary Table 7.

**GUS assay.** Plant samples consisting of the PpMACRO2 fused to EGFP and GUS in the WT background were cultivated on BCD medium, and incubated in 20 µl 1× GUS solution. The histochemical GUS activity was detected using GUS stain kit (Real-Times, China). The GUS reaction mix consisted of the following: 50 mmol $l^{-1}$ potassium ferrocyanide, 50 mmol $l^{-1}$ potassium ferricyanide, 1 mol $l^{-1}$ sodium phosphate buffer, 0.5 mol $l^{-1}$ sodium EDTA, 10% Triton X-100 and water. A separate solution of X-Gluc (5-bromo-4-chloro-3-indolyl-beta-D-glucuronide) at a concentration of 50 mg X-Gluc $ml^{-1}$ of N-N dimethylformamide was added to the above reaction mix at a ratio of 20 µl of X-Gluc solution to 1 ml of reaction mix. The reaction was performed at 37 °C, and reaction time was depending on the tissues examined. Pigments in plant tissues were removed by absolute ethyl alcohol. The stained tissues were observed and photographed using a microscope (Leica DM5500 B, Germany).

**Observation of GFP.** To observe GFP tagged to PpMACRO2 in the WT background, *P. patens* plant samples, including protoplasts, detached leaves, and gametophores, were prepared and then placed on glass slides. GFP signals of protoplasts, detached leaves, and gametophore shoot apical meristems were observed, and images were obtained using a microscope (Leica DM5500 B,

Germany). The green fluorescence excitation was performed with a 488 nm Argon laser.

**Transcriptome analyses.** Extraction of RNA was performed using a RNeasy Plant Mini Kit (Qiagen) and DNase I (Solarbio) for transcriptome sequencing. RNA-seq libraries were prepared with NEB kit, and sequencing was conducted using a HiSeq X Ten (Illumina) to obtain 150 bp paired-ends. The generated reads were firstly filtered and then mapped onto the reference genome of *P. patens* from Phytozome (https://phytozome.jgi.doe.gov) using HISAT2. The calculation of gene expression levels was performed with StringTie. Finally, differentially expressed genes of *ko* and *OE* lines compared to WT plants were identified using DESeq2.

**Chromosomal ploidy analyses.** The chromosomal ploidy of WT, *ko* and *OE* lines was analyzed, respectively, using BD FacsCalibur (USA) flow cytometer. Protonemata of *P. patens* were treated and incubated with DNA fluorochrome propidium iodide (PI) and the relative fluorescence of the stained nuclei was then measured. The cytometer was equipped with an argon ion laser operating at 488 nm. The PI fluorescence was collected by 620 nm fluorescence-2 (FL2) filter. Parameters for data acquisition were kept constant for all samples. Sample flow rate was set at about 100 nuclei/s and at least 6000 nuclei were acquired for each sample. The results acquired were later analyzed using Cell Quest software. Densely gathered nuclei region in dot plot was gated and considered for final analysis to avoid unwanted counts. The average of coefficient of variation values (CV) for G1 peaks was used to evaluate the results. The results with CV <5% were considered as reliable. Histograms were analyzed using Modifit 3.0 software.

**Reporting Summary.** Further information on research design is available in the Nature Research Reporting Summary linked to this article.

## Data availability

RNA-seq data generated as part of the study have been deposited to the NCBI SRA database under the BioProject accession PRJNA615867 [https://www.ncbi.nlm.nih.gov/bioproject/615867]. The source data underlying Figs. 2–6, and Supplementary Figs 3, 8, 9b–d, 10b–d, 13–15 are provided as a Source Data file. Any other data supporting the findings of this study are available from the corresponding authors upon request.

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

## Acknowledgements

We thank Stefan Rensing and Aizhong Liu for comments and suggestions to improve the manuscript, Jianqiang Wu for providing pM999 vector, Mitsuyasu Hasebe for providing pTN182 and pPOG1 vectors, Bojian Zhong for DNA samples of *Spirogyra sp*, and Yanxia Jia for chromosomal ploidy analyses. This work is funded in part by CAS Light of West China, National Natural Science Foundation of China (31970248), and the Second Tibetan Plateau Scientific Expedition and Research (STEP) Program (2019QZKK0502).

## Author contributions

S.W. and J.H. conceived and designed the study. S.W., Y.G., Q.W., J.Z., G.S., X.H., and J.H. performed experiments and data analyses. M.P.R. and H.S. contributed to data interpretation and manuscript writing. S.W. and J.H. wrote the manuscript.

## Competing interests

The authors declare no competing interests.
