## [Peer Review File · Nature Communications]

Reviewers' comments:

Reviewer #1 (Remarks to the Author):

The authors describe the functional characterisation of the PpMACRO2 gene in *Physcomitrella patens*, which bears a resemblance to those found in mycorrhizae-like fungi. The authors take a reverse genetics approach to link PpMACRO2 function to stem cell regulation and 3D growth processes. Firstly, I would like to remark that the developmental phenotypes described in this paper are of great interest and are very significant. However, careful conclusions must be made with regards to links to stem cell/3D growth regulation. My comments:

Introduction

The introduction begins with an outline of the morphological features developed by plants during terrestrialisation. The paper then steers towards the 3D growth literature, particularly those relating to the APB TFs, DEK1 and CLAVATA. Firstly, a review is cited for DEK1 and not the primary literature (which comprises numerous papers in *Physcomitrella patens*). At this point, I have no clue why the authors have opted to study PpMACRO2? The link to 3D growth becomes even more confusing when we subsequently discover that these proteins are not encoded within the genomes of vascular plants (where 3D growth and indeed stem cell regulation processes are present).

Results

Supplementary Fig S1a – Spirogyra information is superfluous. Supplementary Fig S1b+c looks to be a direct repeat of Figure 1. My opinion is that Fig. 1 should comprise diagrams showing exon-intron structure of PpMACRO2 (sequenced version) and location of domains in PpMACRO2 protein. In my experience, it isn't uncommon for Phytozome gene models to be incorrect. The alignment is valid and should remain in Fig. 1. I feel that the phylogeny should be moved to Supplementary Fig 1.

Description of ko and OE lines:

- Supplementary Figure S5 – 'only minimal expression detected'. Do the authors mean 'no expression'? It seems strange that you would detect a transcript in a full deletion mutant. Southern blotting results should be shown – this is standard for gene targeting experiments in *Physcomitrella patens*.
- Supplementary Fig. 7 superfluous
- Supplementary Fig S8 and S9: Really significant results that I feel have a place in the main figure. These results also demonstrate that the ko and OE of this gene has an overall effect on development, not just on stem cell regulation and/or 3D growth. These could be combined into a single figure or incorporated into the current Fig. 2 (moving out the GUS/GFP data to Fig. 3).
- Supplementary Fig. S11 – should also appear in the main figures.
- Figure 2/3 – GUS/GFP expression appears more widespread than the authors claim. Expression is not restricted to stem cells. Rather in Figure 2d, expression can be observed throughout the filamentous network. Nevertheless, I do agree that expression looks more enriched in stem cells though. Perhaps a partial quantification of a GFP signal might be needed?
- Figure 4 – gametophore development occurs earlier in OE than WT or ko lines. Perhaps it is simply earlier development of the gametophore initial cells that causes the enlarged gametophore phenotype? It is unclear why the authors are examining detached leaves.
- Figure 5 – it is possible that the altered expression of AP2 genes is an indirect consequence of having fewer or more gametophore initial cells rather PpMACRO2 acting transcriptionally upstream.
- Throughout – within figure legends and the methods section, it is unclear what the error bars refer to (SEM?) and which statistical tests were performed in each case.

For me, the link to 3D growth in particular is somewhat tenuous. I appreciate that the number of gametophore initial cells appears to be affected, but the gametophores have no apparent cell

orientation defects (like the dek1 and CLAVATA mutants). A stem cell focused view seems more appropriate... (see Aoyama et al.).

Methods

- How were the gametophore assays performed? It doesn't look like the starting material was normalised in anyway (OD measurements or similar). How much material was added to the medium? Was the same amount added?
- Components of the GUS solution?
- In general, the methods section lacks sufficient detail.

Reviewer #2 (Remarks to the Author):

In their article "Mycorrhizae-like gene regulates stem cell development and three-dimensional growth in mosses", Wang and colleagues present a thorough wet lab study on the molecular function of the PpMACRO2 protein. I generally think that the authors have done a study of high quality and an impressive amount work. Yet, the bioinformatic analysis should be revised. Further, the evolutionary interpretations have some problems and require additional thought.

From the perspective of an evolutionary biologist, the main problem of the manuscript is that bryophytes--or in this case Physcomitrella in particular--are treated as living representatives of the earliest land plants. This is wrong. The earliest land plants were organisms that lived 500 million years ago. They are extinct. Physcomitrella is as divergent as any plant living today from the last common ancestor of land plants. Read and cite "Reconstructing trait evolution in plant evo-devo studies" (Delaux et al. 2019; Current Biology 29 (21), R1110-R1118).

Furthermore, with the recent analyses of Puttick et al. 2018 (which is cited) the bryophytes are most likely not the earliest diverging land plants. This makes the aforementioned statement even more problematic.

 Hence, sentences like "pathways that are only retained in bryophytes represent part of the genetic toolkit of ancestral land plants." are misleading and should be reconsidered. Anything found in any plant genome--no matter if moss, spruce or barley--might have been part of the genetic toolkit of ancestral land plants.

l. 44: "fluctuation, and exposure to novel pathogens."  this sentence needs a reference. Read and cite "On plant defense signaling networks and early land plant evolution" (2018; Communicative & Integrative Biology 11 (3), 1-14) as well as "Embryophyte stress signaling evolved in the algal progenitors of land plants" (Proceedings of the National Academy of Sciences 115 (15), E3471-E3480) and "Great moments in evolution: the conquest of land by plants" (2018; Current Opinion in Plant Biology 42, 49-54).

"plants during their transition from water to land (4, 5), but also the sequence of evolutionary changes"  No. To shed light on this topic, the algal relatives of land plants, the Charophytes a.k.a. streptophyte algae must be considered. Read and cite "Plant evolution: landmarks on the path to terrestrial life" (de Vries & Archibald, 2018, New Phytologist 217 (4), 1428-1434)

"homologous genes and processes in flowering plants (e.g., Arabidopsis thaliana). This approach of evolutionary development has identified some core components of key pathways conserved throughout land plant evolution, but, on the other hand, has provided little information on the toolkit specific to early land plants"  for inferring this, ALL land plants must be considered--again, no matter if moss, spruce or barley.

"We suggest that PpMACRO2 is involved in histone modification through ADP-ribosylation"  better say "we speculate" since no supportive data was generated here.

I completely miss from the introduction why PpMACRO2 was chosen as a study subject. Don't get me wrong--it is very interesting indeed. However, it is currently unclear / not justified sufficiently why exactly the PpMACRO2 protein was chosen in an evolutionary context. Therefore, the main motivation is lacking from the introductory part and this hampers the pursuit of a red thread, i.e. being properly guided through the rationale of the article.

Why was only *Intefilum* and not the genome of *Klebsormidium nitens* (back then called *K. flaccidum*) which was published by Hori et al., 2014 in *Nature Communications*? Further, the authors cite the genome paper of Cheng et al. (2019) on *Mesotaenium* and *Spirogloea*. Yet, these genome data are not used by the authors here in this study. This is confusing. Also, *Mesostigma viride* should be queried, which was published by Liang et al. (2019) in *Advanced Science*. Further, why were the many (including only recently published) phylo diverse land plant genomes not used?--such as that of the water fern *Azolla* (see Li et al., 2018; *Nature Plants*) Further, the authors say that MACRO2 is in few other eukaryotes, but it is not clear whether all of these appear in the phylogenies. Further the labeling should be improved. E.g., the glaucophyte sequence can be easily missed in the mass of bacterial sequences. Finally, MACRO2 is also in PFAM database found in eumetazoa.

The authors should seriously consider using an HMM profiling like a pHMMER to screen for MACRO2 in a more robust fashion across eukaryotic genomes / proteomes. Were all of the results included in the phylogenetic analyses? They should be, but currently this is unclear.

Please also provide a table in the supplement for all the hits. And an additional summary table (i.e. how many hits per phylum).

There are some ambiguities regarding the datasets used for the bioinformatic analyses. The M&M does not mention all the databases (e.g. the NCBI EST data) that are mentioned in the results section.

"which only contain transcriptomic data, yielded hits from green algal species, no hit could be identified from any complete genome of green algae that was generated from axenic cultures."  these hits must be specified. The authors should use phylogenetic analyses to try to resolve this. If the gene trees somewhat match the species trees, then these could be meaningful sequences and not just contaminations (such as viral seqs).

Line 133: why not refer to Figure 2C as well here?

"This evidence indicates that PpMACRO2 promotes cell reprogramming"  Please elaborate and explain more why this is your conclusion.

"PpMACRO2 regulates epigenetic modification"  there is not enough support for regulation. Currently a gene expression PATTERN is observed. Please tone this down.

l.214: "genes related to epigenetic modification"  Please explain in the text, not only in the figure, what these epigenetic-associated genes are.

Figure 5: Please provide additional, more meaningful, labels for all the genes in the heatmap. The Pp gene numbers are a good resource but have little informational value by themselves.

In the discussion, many issues regarding the plant terrestrialization concepts / inferences regarding the early land plants that were problematic in the introduction also come to bear here.

l.239/240: "plants and other eukaryotes" this is a bit bold, considering that this study was only

done on *P. patens*.

Lines 242 to 247 should be moved to the results section.

The network analysis is not properly explained in the material and methods section. Please rectify.

Lines 267 to 286: This is all very confusing and needs to be streamlined. What is the direct connection between MACRO2 and facing terrestrial stressors?? Please elaborate.

The context of LGT versus differential loss is especially difficult in eukaryotes; for some stimulating discussion on this, read and cite doi: [10.1002/bies.201700115](https://doi.org/10.1002/bies.201700115) and doi.org/10.1002/bies.201700242

Figure 6 is too cryptic and the figure legend does not help.

Reviewer #3 (Remarks to the Author):

The authors report a novel macro2 domain gene in *Physcomitrella patens* (Pp) an early nonvascular land plant. Macro2 domain genes regulate epigenetic modification, stem cell function, cell reprogramming and other processes in different organisms including plants. This macro2 domain gene (PpMACRO2) is likely derived from Mucoromycota, a mycorrhizae-like fungi found in the earliest land plants.

Sequence comparisons leave little doubt as to the similarity between Pp and fungi.

Fig S 4 not needed and can be briefly stated.

Gene knockout and over-expression results of PpMACRO2 in Pp significantly change the number and size of gametophores, the dominant form of three-dimensional growth in mosses. The authors employ the expected methods to demonstrate these results, so I have no doubts as to their competence or suggestions about their procedures.

Given the strong expression of PpMACRO2 in apical stem cells of both protonemata and gametophores support the role of PpMACRO2 in epigenetic modification, stem cell development, and related process

RNA Seq data supports role in stem cell development.

Major changes in this revision

We are grateful to three reviewers for their detailed and insightful comments and suggestions. We have carefully revised the manuscript based on these comments and suggestions. The following includes some of the major changes in this revision:

1. More experimental details have been added to the manuscript. These include GUS reaction mix, protein subcellular localization, vector construction of knockout and over-expression, plant materials and culture conditions, protein-protein interaction network analyses etc.
2. Southern blotting and GFP intensity quantification were performed, and results have been added to the Supplementary Information (Supplementary Figures 9-10, 13)
3. Figures 1-3 have been reorganized. Some of the original supplementary figures have been moved to Figures 1-3 in the main text.
4. Link between PpMACRO2 to 3D growth has been dropped. The reference to 3D growth has been kept as minimal as possible.
5. Confusions related to bryophytes and early land plant evolution have been clarified. We specifically indicate in this revision that bryophytes, other than their important systematic position, retained many structures and functions, including genes and pathways, that were inherited from early land plants. Some of these genes and pathways might have been lost over the course of land plant evolution, but they are still found in bryophytes. These ancestral genes and pathways, together with those inherited from charophytes, may not only provide critical insights into the adaptive strategies of plants during their transition from water to land, but also the sequence of evolutionary changes that occurred when land plants became increasingly complex, structurally and physiologically (2nd paragraph of Introduction). Additional changes were also made in Discussion and other parts of the manuscript.
6. Taxonomic distribution results from BLAST and pHMMER (Supplementary Figures 1 and 2), as well as alignment files (part of additional source data file) have been added.
7. Other changes were made based on the comments and suggestions by reviewers, as well as to conform to the format requirements of Nature Communications.

Responses to comments by Reviewer #1

1. The authors describe the functional characterisation of the PpMACRO2 gene in *Physcomitrella patens*, which bears a resemblance to those found in mycorrhizae-like fungi. The authors take a reverse genetics approach to link PpMACRO2 function to stem cell regulation and 3D growth processes. Firstly, I would like to remark that the developmental phenotypes described in this paper are of great interest and are very significant. However, careful conclusions must be made with regards to links to stem cell/3D growth regulation.

We are truly grateful to this reviewer for his/her detailed comments and suggestions. We are very glad that the reviewer found the phenotypes significant and highly interesting. We have carefully revised the manuscript based on the comments and suggestions by the reviewer. In particular, the potential link of PpMACRO2 to 3D growth has been dropped. Additional details on experiments have also been added to this revised manuscript (Please see “Major changes in this revision” above).

2. The introduction begins with an outline of the morphological features developed by plants during terrestrialisation. The paper then steers towards the 3D growth literature, particularly those relating to the APB TFs, DEK1 and CLAVATA. Firstly, a review is cited for DEK1 and not the primary literature (which comprises numerous papers in *Physcomitrella patens*). At this point, I have no clue why the authors have opted to study PpMACRO2? The link to 3D growth becomes even more confusing when we subsequently discover that these proteins are not encoded within the genomes of vascular plants (where 3D growth and indeed stem cell regulation processes are present).

Thanks for this comment. Perroud et al. 2018 New Phytologist paper on DEK1 has been cited in this revision.

We apologize for the confusion in the Introduction. Our main argument is that, although conserved core components of pathways are important, other genes that were inherited from the ancestral land plant are also valuable for understanding the early evolution of land plants. These other genes represent part of the gene repertoire of early land plants and were important for early land plant evolution. Some of these genes have been secondarily lost from vascular plants or flowering plants but are still retained in bryophytes. PpMACRO2 is just an example in this regard. In the Discussion section, we also include a paragraph on secondary loss of genes from vascular or seed plants.

Specifically regarding the absence (or secondary loss) of PpMACRO2 from vascular plants, we reason that although stem cells and 3D growth are conserved in all land plants, the underlying genetic mechanisms in early-diverging lineages and flowering plants might not be exactly the same. To understand the early evolution of land plants, not only should the core components be investigated, other related genes retained from early land plants also need to be studied.

To clear the confusions, we moved the description of macrodomains and macro2 to the Introduction section in this revision. We also indicated that *MACRO2* gene was likely acquired by the ancestral land plant from mycorrhizae-like fungi and was secondarily lost from vascular plants. Hopefully this may not only provide a logical link to the discussion of ancestral genes in the paragraph immediately above, but also explain why PpMACRO2 is studied (lines 70-78).

3. Supplementary Fig S1a – *Spirogyra* information is superfluous. Supplementary Fig S1b+c looks to be a direct repeat of Figure 1. My opinion is that Fig. 1 should comprise diagrams showing exon-intron structure of PpMACRO2 (sequenced version) and location of domains in PpMACRO2 protein. In my experience, it isn't uncommon for Phytozome gene models to be incorrect. The alignment is valid and should remain in Fig. 1. I feel that the phylogeny should be moved to Supplementary Fig 1.

Thanks for this suggestion. Changes to Figure 1 have been made according to the suggestions of the reviewer.

The molecular phylogeny of original Figure 1 has been moved to Supplementary Information in this revision (now Supplementary Figure S4). The original Supplementary Figure S1 is also kept in Supplementary Information. These two phylogenies are indeed very similar, and both figures support the relatedness of fungal and land plant *MACRO2* sequences. The original Figure S1 contains additional green algal sequences, which we suspect are contaminations, and shows a more complex evolutionary history of PpMACRO2 (i.e., the origin of PpMACRO2). Given the controversies on horizontal gene transfer in eukaryotes, we think it is important to present both phylogenies so that readers may also make their own judgements.

4. Supplementary Figure S5 – ‘only minimal expression detected’. Do the authors mean ‘no expression’? It seems strange that you would detect a transcript in a full deletion mutant. Southern blotting results should be shown – this is standard for gene targeting experiments in *Physcomitrella patens*.

Southern blotting results have been added to the Supplementary Information in this revision (Supplementary Figure 9d and Supplementary Figure 10d). No expression was detected.

5. Supplementary Fig. 7 superfluous

We agree with the reviewer that this figure may not be entirely necessary. However, given the policy of Nature Communications on source data, we sense that the figure will need to be either included in the Supplementary Information or uploaded as separate file online. It may be more useful to leave the figure here so that readers can easily access it.

6. Supplementary Fig S8 and S9: Really significant results that I feel have a place in the main figure. These results also demonstrate that the ko and OE of this gene has an overall effect on development, not just on stem cell regulation and/or 3D growth. These could be combined into a single figure or incorporated into the current Fig. 2 (moving out the GUS/GFP data to Fig. 3).

Thanks again for this suggestion. In this revision, we split the original Figure 2 into two figures: Figure 2 and Figure 3. Original Figure S8 is now part of current Figure 2, which compares protonemata and gametophore developments (cell length and gametophore height) for individual wide-type and mutant plants. For original Figure S9, the statistical results have now been moved to the new Figure 2; the image of individual gametophores is redundant and thus not retained.

The new Figure 3 mostly includes data on the number of gametophores, including original Supplementary Figure S11. GUS/GFP data are also retained in the new Figure 3 to indicate that PpMACRO2 expression is consistent with the changes in gametophore number.

7. Supplementary Fig. S11 – should also appear in the main figures.

This figure has been moved to the new Figure 3 (see above)

8. Figure 2/3 – GUS/GFP expression appears more widespread than the authors claim. Expression is not restricted to stem cells. Rather in Figure 2d, expression can be observed throughout the filamentous network. Nevertheless, I do agree that expression looks more enriched in stem cells though. Perhaps a partial quantification of a GFP signal might be needed?

We have added a figure about GFP intensity to the Supplementary Information (Supplementary Figure 13).

9. Figure 4 – gametophore development occurs earlier in OE than WT or ko lines. Perhaps it is simply earlier development of the gametophore initial cells that causes the enlarged gametophore phenotype? It is unclear why the authors are examining detached leaves.

Thanks for this comment. The gametophore development does occur earlier in OE lines. Gametophores in OE lines are smaller, but their number is higher. We speculate that more cells (including gametophore initial cells) were generated, which results in more gametophores in OE plants. As such, the total energy might be spent on more gametophores, with each having a smaller share. Nevertheless, it is an interesting phenotype that merits further detailed investigations.

The detached leaves were used to understand the role of PpMACRO2 in cell reprogramming and tissue regeneration. This was inspired by Cove and Knight 1993 review [1] and Mosquana et al. 2009 work on Polycomb protein FIE [2]. Both FIF and PpMACRO2 are involved in stem cell functions and multiple developmental processes.

10. Figure 5 – it is possible that the altered expression of AP2 genes is an indirect consequence of having fewer or more gametophore initial cells rather PpMACRO2 acting transcriptionally upstream.

Thanks for this comment. It is possible that the number of gametophore initial cells affects the expression level of AP2 genes. However, in Aoyama et al. 2012 Development paper (AP2-type transcription factors determine stem cell identity in the moss *Physcomitrella patens*), it is reported that four AP2-type transcription factors are indispensable for the formation of gametophore apical cells, and quadruple disruption blocked gametophore formation. Therefore, we reason that AP2 genes affecting the development of gametophore initial cells is a likely scenario.

Following the comments of the reviewer, we have modified Figure 7 in this revision. Dashed lines are used here to indicate that the relationship is less certain.

11. Throughout – within figure legends and the methods section, it is unclear what the error bars refer to (SEM?) and which statistical tests were performed in each case.

Thanks. Information on error bars and statistical tests have been added in this revision.

12. For me, the link to 3D growth in particular is somewhat tenuous. I appreciate that the number of gametophore initial cells appears to be affected, but the gametophores have no apparent cell orientation defects (like the *dek1* and *CLAVATA* mutants). A stem cell focused view seems more appropriate... (see Aoyama et al.).

Thanks again for this comment. We completely agree with this reviewer here. In this revision, the reference to 3D growth has been removed from the title, and is kept as minimal as possible in main text.

13. How were the gametophore assays performed? It doesn't look like the starting material was normalised in anyway (OD measurements or similar). How much material was added to the medium? Was the same amount added?

We added the following information to this revision: *"To count the number of gametophores, 7-day-old protonemata of wild-type, ko and OE lines were harvested and suspended in 1.5-2 ml of sterile water, respectively, and then crushed. 1 ml of the suspension (OD₆₀₀=0.4) was propagated onto BCD medium and grown for about four weeks"* (lines 338-340)

14. Components of the GUS solution?

Components of GUS solution has been provided in this revision: "The GUS reaction mix consisted of the following: 50 mmol l⁻¹ potassium ferrocyanide, 50 mmol l⁻¹ potassium ferricyanide, 1 mol l⁻¹ sodium phosphate buffer, 0.5 mol l⁻¹ sodium EDTA, 10 % Triton X-100 and water. A separate solution of X-Gluc (5-Bromo-4-Chloro-3-Indolyl-Beta-D-Glucuronide) at a concentration of 50 mg X-Gluc ml⁻¹ of N-N dimethyl formamide was added to the above reaction mix at a ratio of 20 µl of X-Gluc solution to 1 ml of reaction mix." (lines 417-422)

15. In general, the methods section lacks sufficient detail.

More details have been added to Methods section. These include GUS reaction mix, protein subcellular localization, vector construction of knockout and over-expression, plant materials and culture conditions, protein-protein interaction network analyses etc.

Responses to comments by Reviewer #2

1. In their article "Mycorrhizae-like gene regulates stem cell development and three-dimensional growth in mosses", Wang and colleagues present a thorough wet lab study on the molecular function of the PpMACRO2 protein. I generally think that the authors have done a study of high quality and an impressive amount work. Yet, the bioinformatic analysis should be revised. Further, the evolutionary interpretations have some problems and require additional thought.

We are glad that this reviewer found this study to be of high quality. We are also grateful to this reviewer for his/her detailed comments and suggestions. We have carefully considered these comments and suggestions in this revision. Some major confusions about using bryophytes as model organisms to understand early evolution of land plants has been clarified. Additional details have also been added to this revised manuscript.

2. From the perspective of an evolutionary biologist, the main problem of the manuscript is that bryophytes--or in this case Physcomitrella in particular--are treated as living representatives of the earliest land plants. This is wrong. The earliest land plants were organisms that lived 500 million years ago. They are extinct. Physcomitrella is as divergent as any plant living today from the last common ancestor of land plants. Read and cite "Reconstructing trait evolution in plant evo-devo studies" (Delaux et al. 2019; Current Biology 29 (21), R1110-R1118). Furthermore, with the recent analyses of Puttick et al. 2018 (which is cited) the bryophytes are most likely not the earliest diverging land plants. This makes the aforementioned statement even more problematic.  Hence, sentences like "pathways that are only retained in bryophytes represent part of the genetic toolkit of ancestral land plants." are misleading and should be reconsidered. Anything found in any plant genome--no matter if moss, spruce or barley--might have been part of the genetic toolkit of ancestral land plants.

We thank this reviewer for his/her comments and feel sorry for the confusion here. We wrote that bryophytes are extant members of early-diverging land plants. By no means did we imply that bryophytes

are the same as early land plants. All living organisms are products of a long history of evolution and, as such, they possess traits inherited from their ancestors and those evolved later within themselves.

Regarding the systematic position of bryophytes and land plant phylogeny, “virtually every conceivable hypothesis regarding bryophyte phylogeny has been proposed”[3]. Nevertheless, the general consensus and the most recent studies remain that bryophytes are basal to other groups of land plants [4-8]. That being said, other than their unique systematic position, bryophytes also retain many more structures and functions inherited from the ancestral land plant, compared to other land plant groups [3, 9, 10]. In our original submission, when we wrote pathways that were “retained” in bryophytes, we meant that they were derived from ancestral land plants and continue to be kept in bryophytes. Nevertheless, we do agree with this reviewer that this may cause misunderstanding and needs clarification.

To avoid confusions, we have now rewritten the paragraph into the following: “*Bryophytes (liverworts, hornworts, and mosses) include extant members of early-diverging land plant lineages. In addition to their important systematic position, bryophytes have also retained many structures and functions, including genes and pathways, that were inherited from ancestral land plants. Some of these genes or pathways might have been lost secondarily from other land plants over time, but they are still found in bryophytes. These ancestral genes and pathways, together with those inherited from charophytes (the closest relatives of land plants), may not only provide critical insights into the adaptive strategies of plants during their transition from water to land, but also the sequence of evolutionary changes that occurred when land plants became increasingly complex, both structurally and physiologically.*” (lines 49-58).

3. 1. 44: "fluctuation, and exposure to novel pathogens."  this sentence needs a reference. Read and cite "On plant defense signaling networks and early land plant evolution" (2018; *Communicative & Integrative Biology* 11 (3), 1-14) as well as "Embryophyte stress signaling evolved in the algal progenitors of land plants" (*Proceedings of the National Academy of Sciences* 115 (15), E3471-E3480) and "Great moments in evolution: the conquest of land by plants" (2018; *Current Opinion in Plant Biology* 42, 49-54).

de Vries et al. 2018 *Commun Integr Biol* and Rensing 2018 paper have been cited.

4. "plants during their transition from water to land (4, 5), but also the sequence of evolutionary changes"  No. To shed light on this topic, the algal relatives of land plants, the Charophytes a.k.a. streptophyte algae must be considered. Read and cite "Plant evolution: landmarks on the path to terrestrial life" (de Vries & Archibald, 2018, *New Phytologist* 217 (4), 1428-1434)

Cited.

5. "homologous genes and processes in flowering plants (e.g., *Arabidopsis thaliana*). This approach of evolutionary development has identified some core components of key pathways conserved throughout land plant evolution, but, on the other hand, has provided little information on the toolkit specific to early land plants"  for inferring this, ALL land plants must be considered--again, no matter if moss, spruce or barley.

We agree with the reviewer here in general. Yes, all land plants need to be considered. This is reflected in the “core components of key pathways conserved throughout land plant evolution”. However, since we are discussing structures and functions specific to the adaptation of early land plants, genes and pathways that were inherited from ancestral land plants and are still retained in basal plants should be particularly informative in this regard.

6. "We suggest that PpMACRO2 is involved in histone modification through ADP-ribosylation"  better say "we speculate" since no supportive data was generated here.

Done

7. I completely miss from the introduction why PpMACRO2 was chosen as a study subject. Don't get me

wrong--it is very interesting indeed. However, it is currently unclear / not justified sufficiently why exactly the PpMACRO2 protein was chosen in an evolutionary context. Therefore, the main motivation is lacking from the introductory part and this hampers the pursuit of a red thread, i.e. being properly guided through the rationale of the article.

Thanks for this comment. In this revision, we moved the description of macrodomains and macro2 to the Introduction section. We have also indicated that *PpMACRO2* gene was likely acquired by the ancestral land plant from mycorrhizae-like fungi and that this gene has been secondarily lost from vascular plants but is still retained in bryophytes (lines 75-78). Hopefully this may provide a logical link to the discussion of ancestral genes in the paragraph immediately above.

8. Why was only *Interfilum* and not the genome of *Klebsormidium nitens* (back then called *K. flaccidum*) which was published by Hori et al., 2014 in Nature Communications?

As indicated in the manuscript, we searched the NCBI nonredundant protein sequence database, which includes annotated protein sequences of *Klebsormidium nitens*. In fact, we re-sequenced the *K. nitens* genome to ensure all original annotations were correct. We also sequenced *Interfilum paradoxum*, a close relative of *Klebsormidium*. *Interfilum* is mentioned specifically in the manuscript because it has not been published yet and it is not in any other sequence database.

9. Further, the authors cite the genome paper of Cheng et al. (2019) on *Mesotaenium* and *Spirogloea*. Yet, these genome data are not used by the authors here in this study. This is confusing.

Sorry for this confusion. We did search the above two genomes (we contacted the lead author Shifeng Cheng on this issue before the paper was published). This has been clarified in the current revision. No hits were found from the two genomes (lines 93-103).

10. Also, *Mesostigma viride* should be queried, which was published by Liang et al. (2019) in Advanced Science.

We did search the *Mesostigma* genome generated by two different groups (Liang et al. 2019 Advanced Science and Wang et al. 2019 Nature Plant). No hits were found from the generated genome data. This has been clarified in the current revision (lines 93-103).

11. Further, why were the many (including only recently published) phylodiverse land plant genomes not used?--such as that of the water fern *Azolla* (see Li et al., 2018; Nature Plants)

We did search *Azolla* and *Salvinia* (fernbase.org). No hits were found from the two fern genomes. We also searched the genome of hornwort *Anthoceros angustus* that was published very recently (one of us, Yanlong Guan, was a co-author of that genome paper). This issue has been clarified in this revision (lines 93-103).

12. Further, the authors say that MACRO2 is in few other eukaryotes, but it is not clear whether all of these appear in the phylogenies. Further the labeling should be improved. E.g., the glaucophyte sequence can be easily missed in the mass of bacterial sequences.

Not all hits from other eukaryotes were included. However, we did try our best to sample hits broadly from representative lineages, such as red algae, animals, alveolates and stramenopiles. The macro2 domain is only about 145 aa, and many of the support values on the gene tree are already low. A significantly larger sampling might lead to further deterioration of branch support values.

We have relabeled the other eukaryotic sequences in this revision (Supplementary Figures. 4 and 5). Hopefully this will help readers to locate them easily in the figures.

13. Finally, MACRO2 is also in PFAM database found in eumetazoa.

Yes, hits to eumetazoa were indeed found in our search. In the phylogenetic tree (Figure 1 in the original submission, now Supplementary Figure 4), it was represented by the bivalve *Crassostrea gigas*.

We also included two other metazoan sequences (i.e., the sponge *Amphimedon queenslandica* and the stone coral *Orbicella faveolata*) (Supplementary Figures 4 and 5).

14. The authors should seriously consider using an HMM profiling like a pHMMER to screen for MACRO2 in a more robust fashion across eukaryotic genomes / proteomes. Were all of the results included in the phylogenetic analyses? They should be, but currently this is unclear.

Thanks for pointing out this issue. pHMMER search was performed against Reference Proteome and the result was generally consistent with the BLASTP search, though there were far fewer hits.

The macro2 domain is only about 140-150 aa in length, including gaps and ambiguously aligned regions. We performed numerous rounds of phylogenetic analyses using different samplings and methods. In all analyses, fungal and land plant sequences were found to be most closely related. Nevertheless, with such a short sequence length, it is impossible to construct a comprehensive tree with all the search hits with sufficient support values (BLASTP search of NCBI nr database alone provided over **6,600 hits**). As such, molecular phylogenies with sequences sampled from representative lineages are presented (Supplementary Figures 4 and 5).

In this revision, we added a sentence about the pHMMER search in both Materials and Methods and Results sections (lines 97, 322-323).

15. Please also provide a table in the supplement for all the hits. And an additional summary table (i.e. how many hits per phylum).

Two summary tables from BLASTP against nr and pHMMER search against Reference Proteomes are added (Supplementary Figures S1 and S2). Because of its large size (there were over 6,600 hits from BLASTP against nr), hit table has been uploaded as a separate file online.

16. There are some ambiguities regarding the datasets used for the bioinformatic analyses. The M&M does not mention all the databases (e.g. the NCBI EST data) that are mentioned in the results section.

Thanks for this comment. We have added “and other relevant databases (e.g., phytozome, NCBI dbEST, FernBase)” when describing the data sources used in our analyses (line 321).

17. "which only contain transcriptomic data, yielded hits from green algal species, no hit could be identified from any complete genome of green algae that was generated from axenic cultures."  these hits must be specified. The authors should use phylogenetic analyses to try to resolve this. If the gene trees somewhat match the species trees, than these could be meaningful sequences and not just contaminations (such as viral seqs).

A majority of green algal hits from transcriptomic data are included in Supplementary Figure S5 (Figure S1 in our original submission), which shows a MACRO2 phylogeny with green algal hits. These algal hits are closely related among themselves in general, but not particularly to the land plant sequences. This relationship also is consistent with the multiple sequence alignment, which shows unique residues shared by fungal and land plant sequences. Although the tree might be interpreted differently, multiple lines of evidence strongly suggest that these algal sequences might indeed be contaminations.

1. No hits from annotated green algal genomes. There are over 60 well annotated chlorophyte genomes at NCBI (<https://www.ncbi.nlm.nih.gov/genome/browse/#!/overview/Chlorophyta>). At least seven charophyte genomes (e.g., *Klebsormidium*, *Interfilium*, *Chara*, *Mesotaenium*, *Spirogloea*, *Mesostigma*, and *Chlorokybus*) are also available. This large number of annotated green algal genomes are generally of much better quality, but our search identified no hits in any of them. Phytozome also contains several well annotated green algal genomes, most of which are also deposited in GenBank. On the other hand, OneKP data are known to have numerous contaminations.
2. PCR verification cannot confirm algal hits. Although hits were found from the charophyte *Spirogyra pratensis* in NCBI dbEST database, our genomic PCR using various primer pairs could

not amplify sequences from an unspecified congeneric species *Spirogyra sp.* (Supplementary Figure 3)

3. Viruses are very common in green algae. Not only do some viruses (e.g., prasinoviruses) infect related green algae, their sequences from infected host algae are highly similar as well [11, 12]. Therefore, even if the relationships among sampled sequences mirror those of host algae, there is no guarantee that these sequences are truly from their algal hosts. This issue merits some comprehensive PCR verification.

18. Line 133: why not refer to Figure 2C as well here?

Figures 1-3 have been completely redone. It now refers to Figure 3c and has been cited in this revision (lines 138-140).

19. "This evidence indicates that PpMACRO2 promotes cell reprogramming"  Please elaborate and explain more why this is your conclusion.

We changed to "These data indicate that PpMACRO2 promote cell reprogramming" (line 200). It is the overall evidence, including *ko*, *OE*, and GUS staining and GFP. In particular, Figure 5b clearly shows that, compared to WT, *OE* lines produced more filaments from the detached leaves whereas no filaments were found in *ko* lines.

20. "PpMACRO2 regulates epigenetic modification"  there is not enough support for regulation. Currently a gene expression PATTERN is observed. Please tone this down.

Thanks. The subtitle has been changed to "PpMACRO2 affects epigenetics and transcription factors" in this revision.

21. 1.214: "genes related to epigenetic modification"  Please explain in the text, not only in the figure, what these epigenetic-associated genes are.

Thanks for this suggestion. We moved some of the epigenetic genes to paragraph immediately above (lines 207-211). Additional information for related genes is also provided in Supplementary Table 3 in this revision.

22. Figure 5: Please provide additional, more meaningful, labels for all the genes in the heatmap. The Pp gene numbers are a good resource but have little informational value by themselves.

Some gene names are very long, and it is difficult to fit them in the figure. In this revision, we have added Supplementary Table 3 for additional information of these genes. Because the identifiers are from Phytozome, the name, sequences and other related information of each gene are readily accessible to the *Physcomitrella* research community. We have also added a note in the Figure 6 legends in this regard.

23. In the discussion, many issues regarding the plant terrestrialization concepts / inferences regarding the early land plants that were problematic in the introduction also come to bear here.

To clarify the confusions, we have rephrased the sentences into the following in this revision: "Information on ancestral physiological and developmental pathways in early land plants provides unique insights into the strategies or toolkits adopted by plants during their transition from water to land. Bryophytes retain many features evolved in early land plants and thus provide a living laboratory to understand these pathways and related adaptive strategies[3, 9]" (lines 232-235).

24. 1.239/240: "plants and other eukaryotes" this is a bit bold, considering that this study was only done on *P. patens*.

To our knowledge, this work represents the first functional study on macro2 domain. Macro2 is not only commonly found in mosses and maybe in liverworts, but also in other eukaryotes. We believe this conclusion is reasonable.

25. Lines 242 to 247 should be moved to the results section.

Done (lines 212-213).

26. The network analysis is not properly explained in the material and methods section. Please rectify.

Thanks for this comment. We have added the following sentence in the Materials and Methods section: “*Protein-protein interaction network analyses for PpMACRO2 were performed using STRING database (<https://string-db.org>)*”. (lines 330-331)

27. Lines 267 to 286: This is all very confusing and needs to be streamlined. What is the direct connection between MACRO2 and facing terrestrial stressors?? Please elaborate.

Macrodomains, including macro2 to which PpMACRO2 belongs, are most commonly involved in ADP-ribosylation, which in turn is well known for its important role in plant stress responses.

In this revision, we restructured this paragraph. The discussion on the relationship of macrodomains and plant stress responses now immediately follows the sentence on terrestrial stresses (lines 278-284). Hopefully this makes the discussion easier to follow.

27. The context of LGT versus differential loss is especially difficult in eukaryotes; for some stimulating discussion on this, read and cite doi: 10.1002/bies.201700115 and doi.org/10.1002/bies.201700242.

Cited.

28. Figure 6 is too cryptic and the figure legend does not help.

Thanks for this comment. We have added more details to the figure legends in this revision.

Responses to comments by Reviewer #3

1. The authors report a novel macro2 domain gene in *Physcomitrella patens* (Pp) an early nonvascular land plant. Macro2 domain genes regulate epigenetic modification, stem cell function, cell reprogramming and other processes in different organisms including plants. This macro2 domain gene (PpMACRO2) is likely derived from Mucoromycota, a mycorrhizae-like fungi found in the earliest land plants. Sequence comparisons leave little doubt as to the similarity between Pp and fungi.

We thank this reviewer for his/her comments, particularly on the relationship between bryophytes and fungal MACRO2 sequences. The sequence similarity between bryophytes and fungi is intriguing. There are other genes that appear to be of fungal ancestry in mosses. We speculate that these genes might have been transferred between fungi and mosses and are currently working on their functions. Hopefully the studies on these genes might provide a better understanding of organismal interaction and genetic integration between the two groups.

2. Fig S 4 not needed and can be briefly stated.

We agree with the reviewer that this figure may be briefly stated in the text. However, given the policy of Nature Communications on source data, we sense that the figure will need to be either included in the Supplementary Information or uploaded as separate file online. It may be more useful to leave the figure here so that readers can easily access it.

3. Gene knockout and over-expression results of PpMACRO2 in Pp significantly change the number and size of gametophores, the dominant form of three-dimensional growth in mosses. The authors employ the expected methods to demonstrate these results, so I have no doubts as to their competence or suggestions about their procedures.

We thank this reviewer tremendously for his/her encouragement and confidence in our work.

4. Given the strong expression of PpMACRO2 in apical stem cells of both protonemata and

gametophores support the role of PpMACRO2 in epigenetic modification, stem cell development, and related process RNA Seq data supports role in stem cell development.

Thanks again for this comment. The *ko* and *OE* phenotypes of *PpMACRO2* are indeed quite significant. Although the link of *PpMACRO2* to epigenetic modification appears to be clear, there is a lot to be done to understand the detailed mechanisms. We are currently performing additional experiments and analyses in this regard.

References Cited

1. Cove, D.J., and Knight, C.D. (1993). The Moss *Physcomitrella patens*, a Model System with Potential for the Study of Plant Reproduction. *The Plant cell* 5, 1483-1488.
2. Mosquna, A., Katz, A., Decker, E.L., Rensing, S.A., Reski, R., and Ohad, N. (2009). Regulation of stem cell maintenance by the Polycomb protein FIE has been conserved during land plant evolution. *Development (Cambridge, England)* 136, 2433-2444.
3. Renzaglia, K.S., Duff, R.J.T., Nickrent, D.L., and Garbary, D.J. (2000). Vegetative and reproductive innovations of early land plants: implications for a unified phylogeny. *Philosophical transactions of the Royal Society of London. Series B, Biological sciences* 355, 769-793.
4. Rensing, S.A. (2018). Plant Evolution: Phylogenetic Relationships between the Earliest Land Plants. *Current biology : CB* 28, R210-r213.
5. Nishiyama, T., Sakayama, H., de Vries, J., Buschmann, H., Saint-Marcoux, D., Ullrich, K.K., Haas, F.B., Vanderstraeten, L., Becker, D., Lang, D., et al. (2018). The Chara Genome: Secondary Complexity and Implications for Plant Terrestrialization. *Cell* 174, 448-464.e424.
6. Wang, S., Li, L., Li, H., Sahu, S.K., Wang, H., Xu, Y., Xian, W., Song, B., Liang, H., Cheng, S., et al. (2019). Genomes of early-diverging streptophyte algae shed light on plant terrestrialization. *Nat Plants*.
7. Cheng, S., Xian, W., Fu, Y., Marin, B., Keller, J., Wu, T., Sun, W., Li, X., Xu, Y., Zhang, Y., et al. (2019). Genomes of Subaerial Zygnematophyceae Provide Insights into Land Plant Evolution. *Cell* 179, 1057-1067.e1014.
8. Zhang, J., Fu, X.X., Li, R.Q., Zhao, X., Liu, Y., Li, M.H., Zwaenepoel, A., Ma, H., Goffinet, B., Guan, Y.L., et al. (2020). The hornwort genome and early land plant evolution. *Nat Plants* 6, 107-118.
9. Ligrone, R., Duckett, J.G., and Renzaglia, K.S. (2012). Major transitions in the evolution of early land plants: a bryological perspective. *Ann Bot* 109, 851-871.
10. Bowman, J.L., Kohchi, T., Yamato, K.T., Jenkins, J., Shu, S., Ishizaki, K., Yamaoka, S., Nishihama, R., Nakamura, Y., Berger, F., et al. (2017). Insights into Land Plant Evolution Garnered from the *Marchantia polymorpha* Genome. *Cell* 171, 287-304.e215.
11. Weynberg, K.D., Allen, M.J., and Wilson, W.H. (2017). Marine Prasinoviruses and Their Tiny Plankton Hosts: A Review. *Viruses* 9.
12. Moreau, H., Piganeau, G., Desdevises, Y., Cooke, R., Derelle, E., and Grimsley, N. (2010). Marine prasinovirus genomes show low evolutionary divergence and acquisition of protein metabolism genes by horizontal gene transfer. *J Virol* 84, 12555-12563.

REVIEWERS' COMMENTS:

Reviewer #1 (Remarks to the Author):

The authors have taken the reviewers' comments seriously, and have given them good consideration. I am content with the revisions made.

Reviewer #2 (Remarks to the Author):

The authors have fully and satisfactorily tackled all of my previous comments. I do not have any further remarks.